# Dynamical symmetry indicators for Floquet crystals

Jiabin Yu [1✉], Rui-Xing Zhang [1,2] & Zhi-Da Song [3]

Various exotic topological phases of Floquet systems have been shown to arise from crystalline symmetries. Yet, a general theory for Floquet topology that is applicable to all crystalline symmetry groups is still in need. In this work, we propose such a theory for (effectively) non-interacting Floquet crystals. We first introduce quotient winding data to classify the dynamics of the Floquet crystals with equivalent symmetry data, and then construct dynamical symmetry indicators (DSIs) to sufficiently indicate the inherently dynamical Floquet crystals. The DSI and quotient winding data, as well as the symmetry data, are all computationally efficient since they only involve a small number of Bloch momenta. We demonstrate the high efficiency by computing all elementary DSI sets for all spinless and spinful plane groups using the mathematical theory of monoid, and find a large number of different nontrivial classifications, which contain both first-order and higher-order 2+1D anomalous Floquet topological phases. Using the framework, we further find a new 3+1D anomalous Floquet second-order topological insulator (AFSOTI) phase with anomalous chiral hinge modes.

[1] Condensed Matter Theory Center, Department of Physics, University of Maryland, College Park, MD, USA. [2] Joint Quantum Institute, University of Maryland, College Park, MD, USA. [3] Department of Physics, Princeton University, Princeton, NJ, USA. ✉email: jiabinyu@umd.edu

Crystalline symmetries are crucial in the study of static band topology[1–7], because they can protect and can efficiently indicate exotic topological phases[8–29]. Powerful theories, topological quantum chemistry[30] and symmetry indicators[31,32], have been formulated to systematically characterize the crystalline-symmetry-protected (and crystalline-symmetry-indicated) topological phases, which enabled the prediction of thousands of topologically nontrivial materials in a computationally efficient manner[33–36]. The power of the two theories relies on the following two features. First, the two theories can be formally applied to all crystalline symmetry groups. Second, the topological invariants proposed in the two theories are computationally efficient, as they only involve a small number of high-symmetry momenta[11,32,37–40] instead of the entire first Brillouin zone (1BZ).

Beyond the static paradigm, non-interacting Floquet systems[41–63]—systems with noninteracting time-periodic Hamiltonian—can host anomalous topological phases[64–72] that have no analog in any static systems, such as phases with anomalous chiral edge modes in the absence of nonzero Chern numbers[64,66,72]. Recently, researchers have recognized the important role of crystalline (or space-time) symmetries in protecting driving-induced higher-order topological phases in Floquet systems[73–97], and predicted exotic physical phenomena like anomalous corner modes. In particular, ref. [94] introduces a systematic theoretical framework of classifying and characterizing 2+1D anomalous Floquet higher-order topological phases protected by point group and chiral symmetries.

In the field of Floquet topological phases, there are two (among others) open questions that are fundamentally and practically important. The first one is the topological classification for all crystalline symmetry groups, namely how to efficiently determine whether two generic Floquet crystals with the same crystalline symmetries are topologically equivalent. The second one is how to efficiently determine whether a generic Floquet crystal is in an anomalous phase that has no analog in any static systems. In this work, we refer to such inherently dynamical Floquet crystals as Floquet crystals with obstruction to static limits, in analog to the Wannier obstruction[22,30,31,98] for topologically nontrivial phases of static crystals. Then, the second question can be rephrased as how to efficiently diagnose the obstruction to static limits, which is essential for all the above-mentioned anomalous Floquet topological phenomena.

Unfortunately, there have been few efforts to address the above two open questions in the literature, and the previous related works focused on either specific models or special types of crystalline symmetry groups. There have been no general theory that is applicable to all crystalline symmetry groups in all spatial dimensions. Furthermore, the topological invariants proposed in the previous studies have relatively low computational efficiency, since they either do not have an accessible mathematical expression or typically require the information over the entire 1BZ (or a submanifold with nonzero dimensions). Therefore, a general and computationally efficient theory for Floquet topology that takes crystalline symmetries into account is in need.

In this work, we introduce a general theoretical framework to characterize the topological properties of Floquet crystals, which is applicable to all crystalline symmetry groups in all spatial dimensions (up to three). As a demonstration of our general principle, we focus on non-interacting Floquet crystals in the symmetry class A[3,70]—in the absence of time-reversal, particle-hole and chiral symmetries—because an applied drive can break the time-reversal symmetry, and exact particle-hole and chiral symmetries hardly appear in normal phases of crystals. The brief logic is shown in Fig. 1. We introduce quotient winding data, which, together with the symmetry data[30,31] of the quasi-energy

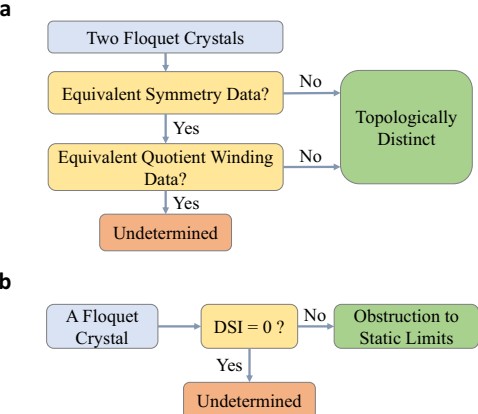

**Fig. 1 Brief summary of main results. a** The flowchart for topologically classifying Floquet crystals based on symmetry data and quotient winding data. **b** The flowchart for using DSI to indicate obstruction to static limits.

bands, provides a topological classification of Floquet crystals (Fig. 1a). In a two-step manner, the symmetry data first provides a coarse classification, which omits the essential information of the dynamics, and the quotient winding data then classifies the dynamics of Floquet crystals with equivalent symmetry data. Based on our classification scheme, we further introduce the concept of DSIs to indicate the obstruction to static limits (Fig. 1b). A nonzero DSI is a sufficient condition for Floquet crystals to have obstruction to static limits. The DSI constructed in this work for Floquet crystals is a dynamical generalization of the symmetry indicator proposed in ref. [31] for static crystals.

Notably, all indices in our theory—including symmetry data, quotient winding data, and thus the DSI—only involve a small number of Bloch momenta in 1BZ, indicating that the evaluation of them is highly computationally efficient or even analytically feasible. As a demonstration of the high efficiency, we provide a table of all elementary DSI sets for all spinless and spinful plane groups. Using specific models, we show that the resultant DSIs can efficiently indicate the nontrivial dynamics in both first-order and higher-order 2+1D anomalous Floquet topological phases. We further apply our framework to the 3+1D inversion-invariant case (space group P$\bar{1}$ or #2), and find a 3+1D Floquet phase with anomalous chiral hinge modes, which is the first 3+1D AFSOTI phase that is solely protected by static crystalline symmetries. It is both the generality and high computational efficiency that make our theory remarkably powerful for prediction of new Floquet topological phases.

## Results

In the following, we will introduce our framework based on symmetry data, quotient winding data, and DSI. We will use a 1+1D inversion-invariant example to illustrate the main idea. Then, we will discuss the DSIs for 2+1D systems. At last, we will introduce the 3+1D AFSOTI phase that we find using DSI. We also briefly describe the general framework in Methods, and the details can be found in Supplementary Notes 2 and 3.

The 1+1D inversion-invariant example that we will use is constructed on a 1D lattice with lattice constant being 1, and each lattice site consists of two orbitals at the same position: one spinless s orbital and one spinless p orbital. As we consider the noninteracting cases, we only care about the single-particle Hilbert space, and the symmetry group $\mathcal{G}$ of interest is spanned by the 1D lattice translations and the inversion symmetry. With bases $|\psi_k\rangle = (|\psi_{k,\mathrm{s}}\rangle, |\psi_{k,\mathrm{p}}\rangle)$, the single-particle Floquet Hamiltonian is represented as $H(k,t)$, where $H(k, t+T) = H(k,t)$ with

$T > 0$ the time period, and $k$ is the Bloch momentum. The corresponding unitary time-evolution matrix $U(k, t)$ can be given by Dyson series. (See Supplementary Note 1 for detailed forms of $H$ and $U$ for this 1+1D example.) Furthermore, the inversion symmetry $\mathcal{P}$ is represented as $\mathcal{P}|\psi_k\rangle = |\psi_{-k}\rangle u_{\mathcal{P}}(k)$ with $u_{\mathcal{P}}(k) = \sigma_z$, where $\sigma$'s are the Pauli matrices. The inversion invariance of the system leads to

$$u_{\mathcal{P}}(k)U(k, t)u_{\mathcal{P}}^{\dagger}(k) = U(-k, t). \tag{1}$$

The quasi-energy spectrum of $U(k, t)$, derived from diagonalizing $U(k, T)$, is important for our later discussion. We plot the quasi-energy spectrum for $U(k, t)$ in Fig. 2a for one set of parameter values, showing two quasi-energy bands in the phase Brillouin zone (PBZ) $[\Phi_k, \Phi_k + 2\pi)$, which are separated by two quasi-energy gaps. (See Supplementary Note 1 for details.) The parameter values used in Fig. 2 give us one specific Floquet system; if we change the parameter values, we would get a new time-evolution matrix $U'(k, t)$, featuring another Floquet system. For this 1+1D example, two Floquet systems are considered to be topologically equivalent if and only if (iff) they are connected by a continuous deformation that preserves the symmetry group $\mathcal{G}$ and both quasi-energy gaps.

In terms of the general terminology discussed in the Methods, we choose both quasi-energy gaps to be the topologically relevant gaps[70,71] for this 1+1D example, and after choosing the relevant gaps, $U(k, t)$ (or $U'(k, t)$) becomes a Floquet gapped unitary (FGU). In general, the relevant gaps for a generic FGU are chosen based on the physics of interest, and one common choice is to choose all quasi-energy gaps to be relevant, as done in this 1+1D example. Topological properties of FGUs are the focus of this work.

**Symmetry data**. As the first step of our topological classification, let us describe the symmetry data for the quasi-energy band structure of the 1+1D FGU $U(k, t)$. Owing to the inversion invariance, the eigenvectors for the quasi-energy bands at an inversion-invariant momentum $k_0 = \Gamma/X$ have definite parities $\alpha = \pm$, as shown in Fig. 2a. For each quasi-energy band $\mathcal{E}_{m,k}$ ($m = 1, 2$), we can count the number of eigenvectors carrying parity $\alpha$ at each $k_0$, denoted by $n_{k_0, \alpha}^m$. As a result, we have a four-component vector for the $m$-th quasi-energy band as

$$A_m = (n_{\Gamma,+}^m, n_{\Gamma,-}^m, n_{X,+}^m, n_{X,-}^m)^T, \tag{2}$$

of which the values can be read out from Fig. 2a as

$$A_1 = (1, 0, 1, 0)^T, \quad A_2 = (0, 1, 0, 1)^T. \tag{3}$$

The symmetry data is the matrix $A$ that has $A_1$ and $A_2$ as its two columns

$$A = (A_1 \, A_2), \tag{4}$$

which clearly only involves two momenta in the 1D 1BZ. The four components of $A_m$ in Eq. (2) are not independent, as they satisfy the following compatibility relation[30,31] $n_{\Gamma,+}^m + n_{\Gamma,-}^m = n_{X,+}^m + n_{X,-}^m$ or equivalently

$$\mathcal{C}A_m = 0 \tag{5}$$

with the compatibility matrix $\mathcal{C}$ being

$$\mathcal{C} = \begin{pmatrix} 1 & 1 & -1 & -1 \end{pmatrix}. \tag{6}$$

The above derivation of symmetry data for the quasi-energy band structure is for a given choice of PBZ (as in Fig. 2a), which is exactly the same as that for a static crystalline system[30,31]. However, the freedom of choosing PBZ for Floquet crystals leads to an additional subtlety in determining the symmetry data, which is absent in dealing with static crystals. As shown in

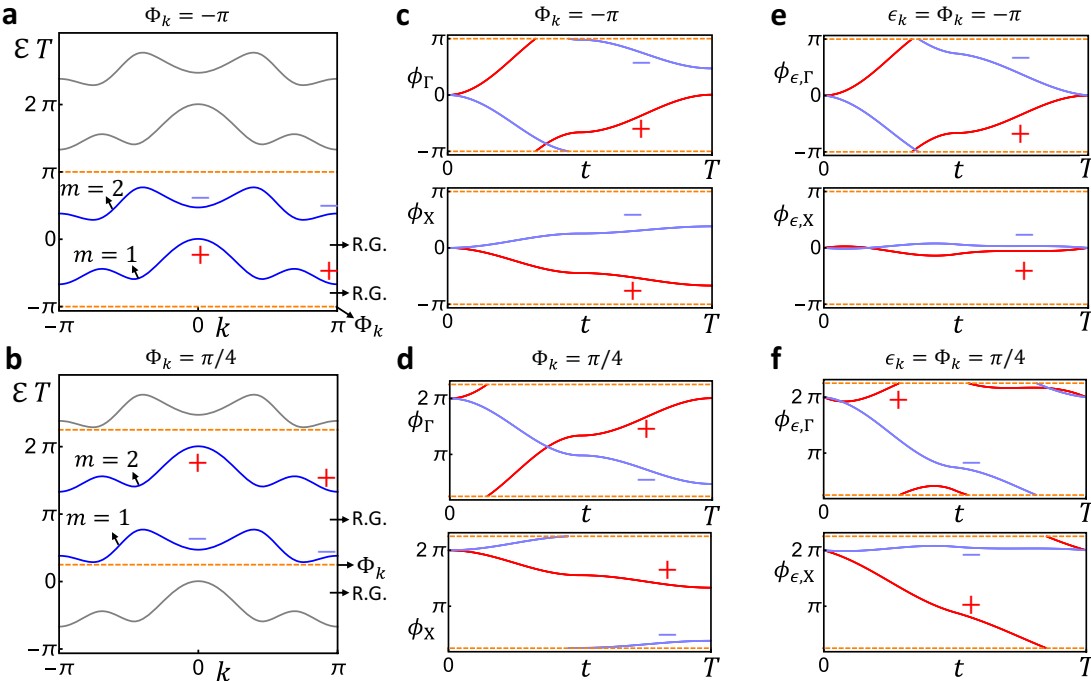

**Fig. 2 Plots of phase and quasi-energy bands for the two-band 1+1D inversion-invariant example.** We choose the PBZ lower bound as $\Phi_k = -\pi$ in **a**, **c**, **e** and as $\Phi_k = \pi/4$ in **b**, **d**, **f**. In all plots, the orange dashed lines mark the boundary of the PBZ, and ± stands for the parity of the eigenvectors at $\Gamma$ ($k = 0$) or $X$ ($k = \pi$). In **a** and **b**, we plot the quasi-energy bands (blue lines) given by $U(k, T)$ in the PBZ, while all gray lines are redundant $2\pi$-copies outside the PBZ. All quasi-energy gaps are relevant for the topological equivalence, and thereby are relevant gaps (labeled by R.G. in the plots). In **c** and **d**, we plot the phase bands at $\Gamma$ and $X$ for the time-evolution matrix $U(k, t)$. In **e** and **f**, we plot the phase bands at $\Gamma$ and $X$ for the return maps $U_{\epsilon=\Phi}(k, t)$.

Fig. 2b, we can legitimately shift the PBZ lower bound to $\Phi_k = \pi/4$, which relabels the quasi-energy bands as $1 \to 2$ and $2 \to 1$, resulting in a new symmetry data $\widetilde{A}$

$$\widetilde{A} = A \begin{pmatrix} 0 & 1 \\ 1 & 0 \end{pmatrix} \text{ for } \Phi_k = \pi/4 . \tag{7}$$

Therefore, the symmetry data of a Floquet crystal depends on the artificial choice of PBZ, which is in contrast to the static case where the symmetry data of a given static crystal is uniquely determined by the Fermi energy.

We remove this artificial PBZ-dependent ambiguity by defining an equivalence among symmetry data of different FGUs. We define two FGUs $U'(k, t)$ and $U(k, t)$ to have equivalent symmetry data iff we can find PBZs to make their symmetry data exactly the same. In practice, we can first pick a PBZ lower bound $\Phi'_k$ for $U'(k, t)$ and get its symmetry data $A'$. Then we check whether $A' = A$ (Eq. (4)) or $A' = \widetilde{A}$ (Eq. (7)); if one of them is true, $U'(k, t)$ and $U(k, t)$ have equivalent symmetry data, otherwise inequivalent. Despite the ambiguity of the symmetry data, whether two FGUs have equivalent symmetry data or not is independent of the artificial PBZ choice. In particular, inequivalent symmetry data infers topological distinction. Therefore, we can perform a topological classification for FGUs solely based on the symmetry data, similar to what we did for static crystals.

**Winding data**. The above symmetry-data-based classification only involves the time-evolution matrix at $t = T$, missing essential information about the quantum dynamics. To classify the dynamics of Floquet crystals with equivalent symmetry data, we will construct the winding data, which contains the dynamical information on the entire time period. A direct visualization of the quantum dynamics for the 1+1D FGU $U(k, t)$ is its phase band spectrum $\phi_{m,k}(t)$ given by directly diagonalizing $U(k, t)$, and we plot the phase bands at two inversion-invariant momenta in Fig. 2(c-d). However, for the construction of the winding data, it turns out to be inconvenient to directly use $U(k, t)$ or phase bands in Fig. 2(c-d), since they are not time-periodic.

It is much more convenient to use the time-periodic return map[68,71] $U_\epsilon(k, t)$ defined as

$$U_\epsilon(k, t) = U(k, t)[U(k, T)]_\epsilon^{-t/T} , \tag{8}$$

where $\epsilon_k$ is the branch cut for the logrithm used in the return map, and throughout this work, we always set the branch cut to be equal to the PBZ lower bound (i.e., $\epsilon = \Phi$) unless specified otherwise. (See the Methods for more details.) As we want to make winding data computationally efficient, we only care about the return map at two inversion-invariant momenta $\Gamma$ and $X$. Since the return map preserves inversion, its eigenvectors at $k_0 = \Gamma/X$ have definite parities, as shown in Fig. 2e for $\Phi_k = -\pi$. Then, we can count the total winding (along $t$) of the phase bands of $U_{\epsilon = \Phi}(k_0, t)$ with parity $\alpha$, resulting in an integer-valued winding number $\nu_{k_0, \alpha}$. We can read out all four quantized winding numbers from Fig. 2e and further group them into a vector

$$V = \left( \nu_{\Gamma,+}, \nu_{\Gamma,-}, \nu_{X,+}, \nu_{X,-} \right)^T = (1, -1, 0, 0)^T , \tag{9}$$

which we refer to as the winding data of the FGU $U(k, t)$ for $\Phi_k = -\pi$. (See more details in the Methods.) Clearly, the winding data only involves two momenta in the 1D 1BZ.

As exemplified by Eq. (9), the four components of the winding data satisfy a compatibility relation

$$\nu_{\Gamma,+} + \nu_{\Gamma,-} = \nu_{X,+} + \nu_{X,-} , \tag{10}$$

since the total winding of all phase bands at each momentum is

the same. As a result, the winding data share the same compatibility relation as that of the symmetry data (Eq. (5))

$$\mathcal{C} V = 0 , \tag{11}$$

meaning that the winding data takes value in the following set

$$\begin{aligned} \{V\} &= \mathbb{Z}^4 \cap \ker \mathcal{C} \\ &= \{ (q_1, q_2, q_3, q_1 + q_2 - q_3)^T | q_1, q_2, q_3 \in \mathbb{Z} \} \approx \mathbb{Z}^3 . \end{aligned} \tag{12}$$

Shifting the PBZ changes the winding data. In this 1+1D example, if we shift the PBZ lower bound from $\Phi_k = -\pi$ to $\Phi_k = \pi/4$, the phase bands of return map become Fig. 2f, and from Fig. 2f, we know the winding data becomes

$$\widetilde{V} = (0, -1, -1, 0)^T = V - A_1 . \tag{13}$$

Unlike the symmetry data, a $2\pi$-shift of the PBZ $\Phi_k \to \Phi_k + 2\pi$ can also change the winding data

$$V \to V - (1, 1, 1, 1)^T , \tag{14}$$

suggesting that the 1+1D FGU $U(k, t)$ can have an infinite number of different winding data, which explicitly depend on the artificial choice of PBZ.

The infinite PBZ dependence of the winding data makes it hard to directly generalize the equivalence among symmetry data (which only has a finite number of variants for a single FGU) to define an equivalence among the winding data, since finding a single proper PBZ among an infinite number of possible choices is not straightforward. Nevertheless, the infinitely many winding data are related by the symmetry data, as shown in Eq. (13) and Eq. (14). This relation inspires us to define the quotient winding data below, in order to resolve the infinity problem.

**Quotient winding data**. In this part, we define the quotient winding data to resolve the infinity issue of the winding data. To have a finite number of different quotient winding data, we define the quotient winding data to be invariant under all PBZ shifts that keep the symmetry data. For the 1+1D FGU $U(k, t)$, all PBZ shifts that keep the symmetry data are (or are equivalent to) the $2\pi n$-shifts of the PBZ, where $n$ labels an arbitrary integer. Then, we define the quotient winding data $V_Q$ as

$$V_Q = V \bmod \bar{A} , \tag{15}$$

where

$$\bar{A} = A_1 + A_2 = (1, 1, 1, 1)^T . \tag{16}$$

As $2\pi n$-shifts of the PBZ can only change $V$ by multiples of $\bar{A}$ according to Eq. (14), $V_Q$ defined in Eq. (15) is indeed invariant under $2\pi n$-shifts of the PBZ, just like the symmetry data. As a result, the 1+1D FGU $U(k, t)$ only has two different quotient winding data derived from the two winding data in Eqs. (9) and (13) as

$$\begin{aligned} V_Q &= V \bmod \bar{A} = (0, -2, -1, -1)^T \text{ for } \Phi_k = -\pi , \\ \widetilde{V}_Q &= \widetilde{V} \bmod \bar{A} = (0, -1, -1, 0)^T \text{ for } \Phi_k = \pi/4 , \end{aligned} \tag{17}$$

which are related by

$$\widetilde{V}_Q = V_Q - A_1 \bmod \bar{A} . \tag{18}$$

(See more details in the Methods.)

To further remove the remaining PBZ-dependent ambiguity, we define an equivalence among quotient winding data of different FGUs. Recall that the quotient winding data is introduced for a classification of FGUs with equivalent symmetry data, since inequivalent symmetry data already infers topological distinction. Let us suppose that the two different 1+1D FGUs $U(k, t)$ and $U'(k, t)$ have equivalent symmetry data, meaning that

we can always pick PBZ choices $\Phi'_k$ and $\Phi_k$ for $U'(k,t)$ and $U(k,t)$, respectively, such that they have exactly the same symmetry data $A' = A$. Then, we check whether the quotient winding data of $U'(k,t)$ for $\Phi'_k$ is the same as that of $U(k,t)$ for $\Phi_k$; if so, $U'(k,t)$ and $U(k,t)$ are defined to have equivalent quotient winding data.

Given two FGUs with equivalent symmetry data, the artificial PBZ choice has no influence on whether they have equivalent quotient winding data or not. In particular, they must have equivalent quotient winding data if they are topologically equivalent, meaning that inequivalent quotient winding data infers topological distinction. Then, as long as the comparison of $V_Q$ is done for the PBZ choices that yield the same symmetry data, quotient winding data provides a topological classification for all FGUs that have equivalent symmetry data to the 1+1D $U(k,t)$. Specifically, the classification is given by $\{V\}$ in Eq. (12) and $\bar{A}$ in Eq. (16) as

$$\{V_Q\} = \{(0, q_2, q_3, q_2 - q_3)^T | q_2, q_3 \in \mathbb{Z}\} \approx \frac{\{V\}}{\bar{A}\mathbb{Z}} \approx \mathbb{Z}^2, \quad (19)$$

where $\bar{A}\mathbb{Z} = \{q\bar{A} = (q, q, q, q)^T | q \in \mathbb{Z}\}$.

Up to now, we have discussed the relative topological classification based on the symmetry data and the quotient winding data, shown in Fig. 1a. Nevertheless, the $(A, V_Q)$-based classification fails to tell which FGU has obstruction to static limits, i.e., topologically distinct from all static FGUs with the same symmetries. (See the Methods for general definitions.) Yet, determining obstruction to static limits is crucial, because it tells whether the Floquet phase of interest has no static analog or equivalently whether it is allowed to have any anomalous dynamical phenomena. For this purpose, we construct the DSI below.

**DSI**. In this part, we construct the DSI for the 1+1D example to efficiently indicate its obstruction to static limits. To determine the obstruction to static limits for the 1+1D FGU $U(k,t)$ with $\mathcal{G}$ spanned by inversion and lattice translation, we only need to consider the $\mathcal{G}$-invariant static FGUs that have symmetry data equivalent to $U(k,t)$, since $U(k,t)$ must be topologically distinct from all other $\mathcal{G}$-invariant static FGUs. Then, we can check whether any winding data of $U(k,t)$ is forbidden in all static FGUs with symmetry data equivalent to $U(k,t)$; if so, then $U(k,t)$ must have obstruction to static limits. Therefore, although we need to use the quotient winding data to give the relative classification, we can directly use winding data to determine the obstruction. (See Supplementary Note 1 for details.)

The DSI is constructed by formalizing the above criterion. To do so, we first derive a set $\{V_{SL}\}$ by winding each quasi-energy band of the 1+1D $U(k,t)$ along time, which reads

$$\{V_{SL}\} = \{q_1 A_1 + q_2 A_2 = (q_1, q_2, q_1, q_2)^T | q_1, q_2 \in \mathbb{Z}\}, \quad (20)$$

where $A_1$ and $A_2$ are two columns of $A$ in Eq. (4). $\{V_{SL}\}$ is invariant under the relabeling of the quasi-energy bands (i.e. $1 \leftrightarrow 2$) due to a PBZ shift, and it contains all winding data of all $\mathcal{G}$-invariant static FGUs that have symmetry data equivalent to $U(k,t)$. (See details in Supplementary Note 1.) Then, according to Eq. (9), the wind data $V$ of $U(k,t)$ for $\Phi_k = -\pi$ satisfies $V \notin \{V_{SL}\}$, since $\nu_{\Gamma,+} - \nu_{X,+} = 1$ for $V$ while $\nu_{\Gamma,+} - \nu_{X,+} = 0$ for all elements in $\{V_{SL}\}$. It means that $V$ cannot exist in any of the static FGUs that have symmetry data equivalent to $U(k,t)$, and thus $U(k,t)$ must have obstruction to static limits.

It turns out that we are allowed to adopt any PBZ choice for $U(k,t)$ to check the above formalized criterion, and we will always get the same result that $U(k,t)$ has the obstruction to static limits, because Eqs. (13) and (14) suggests $U(k,t)$ always has $\nu_{\Gamma,+} -$

$\nu_{X,+} = 1$ regardless of the PBZ choice. We refer to the PBZ-independent $(\nu_{\Gamma,+} - \nu_{X,+})$ as the DSI for $U(k,t)$—as well as for all other FGUs that have symmetry data equivalent to $U(k,t)$. Formally, the DSI is defined to take values from the following set $\mathcal{X}$

$$\mathcal{X} = \frac{\{V\}}{\{V_{SL}\}} \approx \{\nu_{\Gamma,+} - \nu_{X,+} \in \mathbb{Z}\}, \quad (21)$$

where we have used Eqs. (12) and (20). As shown above, the DSI only involves two momenta in the 1BZ, and nonzero DSI means all winding data of $U(k,t)$ are not in $\{V_{SL}\}$, thus sufficiently indicating the obstruction to static limit.

The idea of using quotient group to mod out the trivial systems (though not exclusively), which is used above, was previously used to construct the static symmetry indicator in ref. [31]. The difference between ref. [31] and our work is that the quotient is taken for the symmetry contents (like columns of symmetry data) in the construction of the static symmetry indicator in ref. [31] to characterize the static band topology, while the quotient is taken for the winding data in the construction of the DSI to characterize the periodic quantum dynamics here.

Now we discuss the DSIs for all possible 1+1D inversion-invariant FGUs. To do so, we need to use the Hilbert bases[27,99], which intuitively speaking, are irreducible bases of the symmetry data. (See Methods and Supplementary Note 3 for more details.) For 1+1D inversion-invariant FGUs, there are four Hilbert bases given by four ways of assigning ± parities to Γ/X, which read

$$\begin{aligned} a_1 &= (1, 0, 1, 0)^T, \\ a_2 &= (0, 1, 1, 0)^T, \\ a_3 &= (1, 0, 0, 1)^T, \\ a_4 &= (0, 1, 0, 1)^T. \end{aligned} \quad (22)$$

Then, any column of any symmetry data of any 1+1D inversion-invariant FGU is the linear combination of the four Hilbert bases with non-negative integer coefficients.

Based on the Hilbert bases, symmetry data of FGUs can be split into two types, irreducible and reducible. Specifically, we define a symmetry data of a FGU to be irreducible iff all its columns are Hilbert bases; otherwise reducible. According to the general framework presented in Methods, DSI sets for reducible symmetry data can be constructed from those for irreducible symmetry data. Therefore, in the following, we focus on the DSI sets for irreducible symmetry data.

For a 1+1D inversion-invariant FGU with irreducible symmetry data, all its symmetry data are spanned by a unique set of the Hilbert bases $\{a_j\}$ with $j$ taking $J \leq 4$ different values in $\{1, 2, 3, 4\}$. According to the general framework discussed in the Methods, we can directly obtain the DSI set for the FGU solely based on the set $\{a_j\}$ and the compatibility matrix $\mathcal{C}$ in Eq. (6). For the above 1+1D example $U(k,t)$, the two columns of any symmetry data (Eqs. (4) or (7)) are the Hilbert bases $a_1$ and $a_4$ in Eq. (22), and thus are irreducible. Then, the unique set of the Hilbert bases that span the symmetry data is $\{a_1, a_4\}$, and the DSI set Eq. (21) can be directly derived from $\{a_1, a_4\}$ and $\mathcal{C}$ based on the general framework.

In particular, even if two 1+1D inversion-invariant FGUs have inequivalent symmetry data, they have the same $\mathcal{X}$, as long as their irreducible symmetry data are spanned by the same set of Hilbert bases. This simplification allows us to enumerate all possible DSI sets for irreducible symmetry data by considering all $2^4 - 1$ nontrivial combinations of Hilbert bases. As a result, we obtain two nontrivial DSI sets. One is for the Hilbert basis set $\{a_1, a_4\}$, which is just the above 1+1D example $U(k,t)$, and the DSI is shown in Eq. (21). The other one is for the Hilbert basis set

$\{a_2, a_3\}$, and the DSI set reads

$$\mathcal{X} \approx \{\nu_{\Gamma,+} - \nu_{X,-} \in \mathbb{Z}\}. \tag{23}$$

Besides indicating obstruction to static limit, DSI is also a topological invariant—its different values infer topological distinction for FGUs with equivalent symmetry data. Although the classification given by DSIs is a subset of that given by quotient winding data (like for the above 1+1D example), DSIs have the advantage of being PBZ-independent.

**2+1D DSI**. The above discussion focused on the 1+1D inversion-invariant case. We, in this part, discuss the DSIs for 2+1D systems. Based on the general framework in Methods, we derive the DSI sets for all nontrivial combinations of Hilbert bases for all spinless and spinful 2D plane groups, and list the numbers of nontrivial DSI sets in Tables 1–2. These DSI sets are for 2+1D FGUs with irreducible symmetry data, and serve as the building blocks for all 2+1D DSI sets for plane groups. The nontrivial DSIs in Tables 1–2 can indicate both first-order and higher-order anomalous Floquet topological phases, as discussed below.

For the first-order phase, we focus on the $\mathbb{Z}^3$ set for spinless plane group p2, which is spanned by the 2D lattice translations and the two-fold rotation $C_2$. We explicitly construct a 2+1D p2-invariant spinless model that has nonzero $\mathbb{Z}^3$ DSI (inspired by the quantum-anomalous-Hall-effect model in ref. [2], and find that the model has anomalous chiral edge modes in the absence of nonzero Chern numbers, similar to the first-order anomalous Floquet topological phase in ref. [64]. (See Supplementary Note 4 for details.) Therefore, the $\mathbb{Z}^3$ DSI can indicate first-order anomalous Floquet topological phases. In particular, all components of the DSI take the same values as the winding number $W$ defined in ref. [64] in our specific p2-invariant model, but the evaluation of the former is much more efficient than the latter, since the former only involves four $C_2$-invariant momenta while the latter needs the whole 2D 1BZ. Then, although the winding number $W$ defined in ref. [64] does not rely on any crystalline

symmetries other than lattice translations, our model suggests that in the presence of nontrivial crystalline symmetries, DSIs might efficiently indicate the nontrivial dynamics of the first-order anomalous Floquet topological phases characterized by the $W$ winding number.

For the higher-order phase, we find that the 2+1D anomalous Floquet higher-order topological insulator phase proposed in ref. [86] can be indicated by the $\mathbb{Z}$ DSI of spinful p4mm in Table 2. (See Supplementary Note 5 for details.) In particular, to determine the nontrivial dynamics in the model, the DSI only requires three momenta in the 1BZ, saving us from evaluating the quantized dynamical quadrupole momoent proposed in ref. [86], which involves all momenta in the entire 2D 1BZ.

**3+1D AFSOTI phase**. In this part, we apply our framework to the 3+1D inversion-invariant case (P$\bar{1}$ space group), and predict a new 3+1D AFSOTI phase. We will only present a brief discussion here, and details can be found in Supplementary Note 6.

For P$\bar{1}$, we only need to care about the eight inversion-invariant momenta—$\Gamma(0,0,0)$, $X(\pi,0,0)$, $Y(0,\pi,0)$, $Z(0,0,\pi)$, $V(\pi,\pi,0)$, $U(\pi,0,\pi)$, $T(0,\pi,\pi)$, and $R(\pi,\pi,\pi)$[30]—and we have parities $\pm$ at each inversion-invariant momentum. Then, we choose the winding data to have the form

$$\left( \nu_{\mathbf{K}_1,+}, \nu_{\mathbf{K}_1,-}, ..., \nu_{\mathbf{K}_8,+}, \nu_{\mathbf{K}_8,-} \right)^T \tag{24}$$

with $\mathbf{K}_i = (\Gamma, X, Y, Z, V, U, T, R)_i$, and $\nu$ is the winding number. Replacing $\nu$ by the number of irreducible representations (irreps) labeled by parities in the above expression gives columns of the symmetry data.

P$\bar{1}$ has 256 Hilbert bases, given by assigning $\pm$ to the eight inversion-invariant momenta in the 3D 1BZ, and thus the number of nontrivial combinations of the Hilbert bases is $2^{256} - 1$, which is very large. For simplicity, we only compute the DSI sets for the 32896 combinations that only include one or two Hilbert bases, resulting in $\mathbb{Z}$ (3584), $\mathbb{Z}^2$ (7168), $\mathbb{Z}^3$ (8960),

---

**Table 1 Numbers of Hilbert bases and nontrivial DSIs for all spinless 2D plane groups.**

| P.G. | H.B.N. | Nontrivial DSI Sets |
|---|---|---|
| p1 | 1 | None |
| p2 | 16 | $\mathbb{Z}$ (2980), $\mathbb{Z}^2$ (268), $\mathbb{Z}^3$ (8), $\mathbb{Z}_2$ (666), $\mathbb{Z}_2 \times \mathbb{Z}$ (24), $\mathbb{Z}_3$ (16) |
| pm | 4 | $\mathbb{Z}$ (2) |
| pg | 1 | None |
| cm | 2 | None |
| p2mm | 24 | $\mathbb{Z}$ (1657492), $\mathbb{Z}^2$ (372286), $\mathbb{Z}^3$ (78060), $\mathbb{Z}^4$ (11904), $\mathbb{Z}^5$ (1200), $\mathbb{Z}^6$ (94), $\mathbb{Z}^7$ (4), $\mathbb{Z}_2$ (354594), $\mathbb{Z}_2 \times \mathbb{Z}$ (63296), $\mathbb{Z}_2 \times \mathbb{Z}^2$ (10320), $\mathbb{Z}_2 \times \mathbb{Z}^3$ (1264), $\mathbb{Z}_2 \times \mathbb{Z}^4$ (65), $\mathbb{Z}_3$ (10392), $\mathbb{Z}_3 \times \mathbb{Z}$ (1024), $\mathbb{Z}_3 \times \mathbb{Z}^2$ (112), $\mathbb{Z}_3 \times \mathbb{Z}^3$ (8), $\mathbb{Z}_4$ (3424), $\mathbb{Z}_4 \times \mathbb{Z}$ (16), $\mathbb{Z}_5$ (16), $\mathbb{Z}_6$ (64) |
| p2mg | 6 | $\mathbb{Z}$ (15), $\mathbb{Z}^2$ (3) |
| p2gg | 4 | $\mathbb{Z}$ (2) |
| c2mm | 14 | $\mathbb{Z}$ (3113), $\mathbb{Z}^2$ (686), $\mathbb{Z}^3$ (99), $\mathbb{Z}^4$ (7), $\mathbb{Z}_2$ (476), $\mathbb{Z}_2 \times \mathbb{Z}$ (168), $\mathbb{Z}_2 \times \mathbb{Z}^2$ (56), $\mathbb{Z}_2 \times \mathbb{Z}^3$ (7), $\mathbb{Z}_3$ (12), $\mathbb{Z}_4$ (2) |
| p4 | 32 | $\mathbb{Z}$ (17587274), $\mathbb{Z}^2$ (491020), $\mathbb{Z}^3$ (20760), $\mathbb{Z}^4$ (336), $\mathbb{Z}_2$ (2175362), $\mathbb{Z}_2 \times \mathbb{Z}$ (56952), $\mathbb{Z}_2 \times \mathbb{Z}^2$ (576), $\mathbb{Z}_3$ (27120), $\mathbb{Z}_3 \times \mathbb{Z}$ (384), $\mathbb{Z}_4$ (144) |
| p4mm | 26 | $\mathbb{Z}$ (6044617), $\mathbb{Z}^2$ (859049), $\mathbb{Z}^3$ (116266), $\mathbb{Z}^4$ (11202), $\mathbb{Z}^5$ (597), $\mathbb{Z}^6$ (14), $\mathbb{Z}_2$ (422534), $\mathbb{Z}_2 \times \mathbb{Z}$ (81467), $\mathbb{Z}_2 \times \mathbb{Z}^2$ (11010), $\mathbb{Z}_2 \times \mathbb{Z}^3$ (869), $\mathbb{Z}_2 \times \mathbb{Z}^4$ (22), $\mathbb{Z}_3$ (3200), $\mathbb{Z}_3 \times \mathbb{Z}$ (480), $\mathbb{Z}_3 \times \mathbb{Z}^2$ (56), $\mathbb{Z}_3 \times \mathbb{Z}^3$ (4), $\mathbb{Z}_4$ (2400), $\mathbb{Z}_4 \times \mathbb{Z}$ (450), $\mathbb{Z}_4 \times \mathbb{Z}^2$ (42), $\mathbb{Z}_8$ (8), $\mathbb{Z}_8 \times \mathbb{Z}$ (1) |
| p4gm | 11 | $\mathbb{Z}$ (615), $\mathbb{Z}^2$ (99), $\mathbb{Z}^3$ (7), $\mathbb{Z}_2$ (1) |
| p3 | 27 | $\mathbb{Z}$ (973458), $\mathbb{Z}^2$ (48762), $\mathbb{Z}^3$ (2376), $\mathbb{Z}^4$ (36), $\mathbb{Z}_2$ (201690), $\mathbb{Z}_2 \times \mathbb{Z}$ (4968), $\mathbb{Z}_2 \times \mathbb{Z}^2$ (54), $\mathbb{Z}_3$ (2604), $\mathbb{Z}_4$ (324) |
| p3m1 | 12 | $\mathbb{Z}$ (378), $\mathbb{Z}^2$ (27), $\mathbb{Z}_2$ (360), $\mathbb{Z}_2 \times \mathbb{Z}$ (21), $\mathbb{Z}_4$ (16) |
| p31m | 9 | $\mathbb{Z}$ (148), $\mathbb{Z}^2$ (33), $\mathbb{Z}^3$ (3), $\mathbb{Z}_2$ (3), $\mathbb{Z}_2 \times \mathbb{Z}$ (1) |
| p6 | 36 | $\mathbb{Z}$ (110427458), $\mathbb{Z}^2$ (2196588), $\mathbb{Z}^3$ (68760), $\mathbb{Z}_2$ (16472556), $\mathbb{Z}_2 \times \mathbb{Z}$ (254520), $\mathbb{Z}_3$ (148920) |
| p6mm | 20 | $\mathbb{Z}$ (189005), $\mathbb{Z}^2$ (32809), $\mathbb{Z}^3$ (3301), $\mathbb{Z}^4$ (168), $\mathbb{Z}_2$ (6509), $\mathbb{Z}_2 \times \mathbb{Z}$ (1691), $\mathbb{Z}_2 \times \mathbb{Z}^2$ (172), $\mathbb{Z}_3$ (22) |

Here we only consider the DSIs for FGUs with irreducible symmetry data. P.G. means plane group, and H.B.N. means the number of Hilbert bases for each plane group. In the column for nontrivial DSI sets, None means there are no combinations of Hilbert bases that give nontrivial DSIs, and the number in the bracket is the number of Hilbert-bases combinations that give the DSI set in front of the bracket.

**Table 2 Numbers of Hilbert bases and nontrivial DSIs for all spinful 2D plane groups.**

| P.G. | H.B.N. | Nontrivial DSI Sets |
|---|---|---|
| p1 | 1 | None |
| p2 | 16 | $\mathbb{Z}$ (2980), $\mathbb{Z}^2$ (268), $\mathbb{Z}^3$ (8), $\mathbb{Z}_2$ (666), $\mathbb{Z}_2 \times \mathbb{Z}$ (24), $\mathbb{Z}_3$ (16) |
| pm | 4 | $\mathbb{Z}$ (2) |
| pg | 1 | None |
| cm | 2 | None |
| p2mm | 1 | None |
| p2mg | 6 | $\mathbb{Z}$ (15), $\mathbb{Z}^2$ (3) |
| p2gg | 4 | $\mathbb{Z}$ (2) |
| c2mm | 3 | $\mathbb{Z}$ (1), $\mathbb{Z}_2$ (1) |
| p4 | 32 | $\mathbb{Z}$(17587274), $\mathbb{Z}^2$(491020), $\mathbb{Z}^3$(20760), $\mathbb{Z}^4$(336), $\mathbb{Z}_2$(2175362), $\mathbb{Z}_2 \times \mathbb{Z}$(56952), $\mathbb{Z}_2 \times \mathbb{Z}^2$(576), $\mathbb{Z}_3$(27120), $\mathbb{Z}_3 \times \mathbb{Z}$(384), $\mathbb{Z}_4$(144) |
| p4mm | 4 | $\mathbb{Z}$ (2) |
| p4gm | 8 | $\mathbb{Z}$ (50), $\mathbb{Z}^2$ (4), $\mathbb{Z}_2$ (2) |
| p3 | 27 | $\mathbb{Z}$ (973458), $\mathbb{Z}^2$ (48762), $\mathbb{Z}^3$ (2376), $\mathbb{Z}^4$ (36), $\mathbb{Z}_2$ (201690), $\mathbb{Z}_2 \times \mathbb{Z}$ (4968), $\mathbb{Z}_2 \times \mathbb{Z}^2$ (54), $\mathbb{Z}_3$ (2604), $\mathbb{Z}_4$ (324) |
| p3m1 | 12 | $\mathbb{Z}$ (378), $\mathbb{Z}^2$ (27), $\mathbb{Z}_2$ (360), $\mathbb{Z}_2 \times \mathbb{Z}$ (21), $\mathbb{Z}_4$ (16) |
| p31m | 9 | $\mathbb{Z}$ (148), $\mathbb{Z}^2$ (33), $\mathbb{Z}^3$ (3), $\mathbb{Z}_2$ (3), $\mathbb{Z}_2 \times \mathbb{Z}$ (1) |
| p6 | 36 | $\mathbb{Z}$ (110427458), $\mathbb{Z}^2$ (2196588), $\mathbb{Z}^3$ (68760), $\mathbb{Z}_2$ (16472556), $\mathbb{Z}_2 \times \mathbb{Z}$ (254520), $\mathbb{Z}_3$ (148920) |
| p6mm | 6 | $\mathbb{Z}$ (12) |

Here we only consider the DSIs for FGUs with irreducible symmetry data. P.G. means plane group, and H.B.N. means the number of Hilbert bases for each plane group. In the column for nontrivial DSI sets, None means there are no combinations of Hilbert bases that give nontrivial DSIs, and the number in the bracket is the number of Hilbert-bases combinations that give the DSI set in front of the bracket.

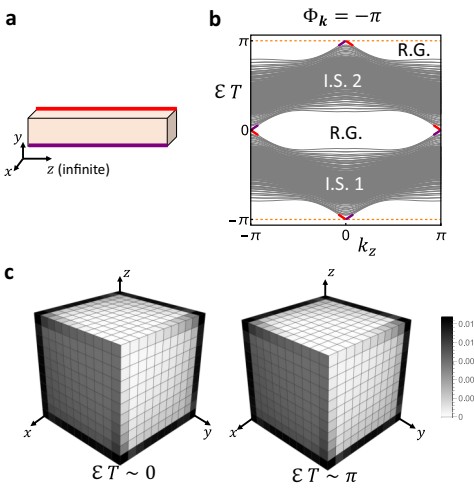

**Fig. 3 The 3+1D AFSOTI with anomalous chiral hinge modes.** In **a**, we show an inversion-preserving configuration of the 3+1D AFSOTI that is finite along $x$, $y$ and infinite along $z$. The purple and red lines indicate the hinges that host anomalous chiral hinge modes. In **b**, we plot the quasi-energy band structure of the 3+1D AFSOTI for the configuration in **a**. $\mathcal{E}$ and $T$ label the quasi-energy and time period, respectively. The gray lines are the bulk bands (as well as the dangling surface bands). I.S. labels isolated set of bulk quasi-energy bands, and each isolated set contains two bulk quasi-energy bands. R.G. stands for the bulk relevant gap, and there are two relevant gaps, one at $\mathcal{E}T = 0$ (called 0-gap) and the other at $\mathcal{E}T = \pi$ (called $\pi$-gap). The purple and red lines mark the anomalous chiral hinge modes localized at the purple and red hinges in **a**, respectively. The orange dashed lines mark the boundary of PBZ. In **c**, we consider an inversion-preserving configuration of the 3+1D AFSOTI that is finite along all three spatial directions, and plot total probability density of the two eigenmodes with quasi-energies closest to 0 or $\pi$ ( mod $2\pi$). Details on the parameter values can be found in Supplementary Note 6.

$\mathbb{Z}^4$ (7168), $\mathbb{Z}^5$ (3584), $\mathbb{Z}^6$ (1024), and, $\mathbb{Z}^7$ (128), where $\mathbb{Z}^n$ labels the DSI set and the number in the bracket labels how many nontrivial combinations of Hilbert bases lead to the DSI set. For concreteness, we in the following focus on the $\mathbb{Z}^7$ DSI set that corresponds to the combination of the following two Hilbert bases

$$\widetilde{a}_1 = (1, 0, 0, 1, 0, 1, 0, 1, 0, 1, 0, 1, 0, 1, 1, 0)^T$$
$$\widetilde{a}_2 = (0, 1, 1, 0, 1, 0, 1, 0, 1, 0, 1, 0, 1, 0, 0, 1)^T, \tag{25}$$

and the DSI is a seven-component vector that reads

$$(\nu_{\Gamma,+} - \nu_{X,-}, \nu_{\Gamma,+} - \nu_{Y,-}, \nu_{\Gamma,+} - \nu_{Z,-}, \nu_{\Gamma,+} - \nu_{V,-},$$
$$\nu_{\Gamma,+} - \nu_{U,-}, \nu_{\Gamma,+} - \nu_{T,-}, \nu_{R,-} - \nu_{\Gamma,-}). \tag{26}$$

To demonstrate the dynamical phase indicated by the $\mathbb{Z}^7$ DSI, we explicitly construct a 3+1D dynamical tight-binding model with P$\bar{1}$ space group on a cubic lattice with the lattice constant being 1. It has four bulk quasi-energy bands, which are split into two isolated sets by two relevant gaps, one 0-gap and one $\pi$-gap (see Fig. 3a, b). According to the Methods, each isolated set (that consists of two bands) corresponds to one column in the symmetry data, resulting in a two-column symmetry data $A = (A_1 \quad A_2)$. Direct calculation shows that $A_1 = 2\widetilde{a}_1$ and $A_2 = 2\widetilde{a}_2$, meaning that the symmetry data is reducible. Nevertheless, the Hilbert basis set that spans the symmetry data is uniquely $\{\widetilde{a}_1, \widetilde{a}_2\}$, and thus the model is characterized by the DSI in Eq. (26), which is evaluated to (2, 2, 2, 2, 2, 2, 2). As a result of the nontrivial dynamics characterized by the nonzero DSI, the system has chiral hinge modes in each bulk relevant gap, as shown in Fig. 3. The chiral hinge modes are anomalous, because the static topological invariant, axion angle, for the inversion-protected chiral hinge modes[100] is zero for both isolated sets of quasi-energy bands according to the symmetry data[11,12].

We emphasize that although hinge modes in 3+1D Floquet insulators were discussed in ref. [77] and ref. [97], our model is fundamentally different from theirs. First, the hinge modes

originate from the time glide symmetry in ref. [77] and from the effective spectral symmetry in ref. [97], both of which are not static crystalline symmetries, while our model is protected by static inversion symmetry. Second, trivial static topology has not been explicitly confirmed for the bulk quasi-energy bands in ref. [77] and ref. [97], while the relevant static topological invariants in our model are confirmed to be trivial. Therefore, our model, which is constructed based on the DSI, is the first AFSOTI solely protected by the static crystalline symmetries.

## Discussion

To summarize, we have established a general and efficient theoretical framework for classifying and characterizing the topological properties of Floquet crystals in the symmetry class A, which is applicable to all crystalline symmetry groups in all spatial dimensions (up to three).

One direct physical implication of the obstruction to static limits is that symmetry breaking or relevant gap closing must appear during any continuous deformation that makes static a Floquet crystal with obstruction. Here the relevant gap closing refers to the closing of the topologically relevant bulk quasi-energy gaps, according to the definition of the topological equivalence discussed in Methods. If all relevant symmetries are preserved during the deformation, the gap closing will occur in at least one of the relevant gaps in the bulk quasi-energy spectrum, and is not required to appear in any irrelevant gaps. Therefore, an experimental test of nonzero DSIs (though not conclusively) would be to observe the quasi-energy gap closing or symmetry breaking as continuously decreasing the driving amplitude to zero while fixing the driving period. Gap closing in quasi-energy spectrum has been observed in experiments like ref. [72].

As for more experimental signatures of our theory, it is worth studying the link between nonzero DSIs and nontrivial boundary signature in the future. A promising direction is p2 plane group, for which the DSI is very likely to contain the information of chiral edge modes. The intuition is based on the fact that the difference in winding data for different PBZ choices consists of the symmetry contents of quasi-energy bands, which have a mod-2 relation to the Chern number[11,12]. The 2+1D p2-invariant model presented above also suggests a relation between the DSI and the winding number defined in ref. [64], where the latter has a correspondence to the chiral edge modes. Furthermore, the 3+1D AFSOTI presented above suggests that the 3+1D DSI might indicate the anomalous chiral hinge modes in certain space groups, perhaps related to a dynamical generalization of the static bulk-boundary correspondence between the axion angle and the static chiral hinge modes[100].

As the symmetry-representation theories for static crystals inspired the proposal of fragile topology[22,23] that is beyond the K-theory classification[101], another interesting direction is to generalize the concept of fragile topology to Floquet crystals[102]. Besides, generalizing our theoretical frame work to Floquet crystals with time-reversal, particle-hole, or chiral symmetries is another interesting direction, since it would help identify exotic physical phenomena like anomalous boundary Majorana modes protected by particle-hole symmetries[46,103–106]. As our framework focuses on operators, it is interesting to ask whether it is possible to formalize an equivalent state-based formalism[63]. Similar to the symmetry-representation theories[30,31] for static crystals, our classification is not necessarily complete, since two Floquet crystals with equivalent symmetry and quotient winding data might still be topologically distinct, and the obstruction to static limits might still occur for zero DSI (Fig. 1). Thereby, the complete topological classification for static and Floquet crystals is a meaningful future direction. Note added in

proof: Recently, we noticed ref. [107], which proposed to classify Floquet topological phases by using dynamical symmetry inversion points.

## Methods

**Basic definitions**. In this part, we list the basic definitions used in this work. Details can be found in Supplementary Note 2.

A Floquet crystal is defined to be a time-evolution operator $\hat{U}(t)$ equipped with a time period $T$, a relevant gap choice, and a crystalline symmetry group $\mathcal{G}$, which is in short denoted by $\hat{U}(t)$. In the definition of a Floquet crystal, we have implied that $\hat{U}(t)$ is unitary and its matrix representation for any bases is continuous. A FGU is defined to be a time-evolution matrix $U(\mathbf{k}, t)$ equipped with a time period $T$, a relevant gap choice, a crystalline symmetry group $\mathcal{G}$, and a symmetry representation $u_g(\mathbf{k})$, which is in short denoted by $U(\mathbf{k}, t)$. Here $\mathbf{k}$ is the momentum. In the definition of a FGU, we have implied that $U(\mathbf{k}, t)$ and $u_g(\mathbf{k})$ are unitary, continuous (smooth for $u_g(\mathbf{k})$), and invariant under the shift of $\mathbf{k}$ by reciprocal lattice vectors. By choosing bases for a Floquet crystal, we naturally get a FGU with the same time period, relevant gaps and crystalline symmetry group as the Floquet crystal. FGUs given by the same Floquet crystal with different choices of bases are related by gauge transformations.

Suppose we have two FGUs $U(\mathbf{k}, t)$ (with $T$, relevant gaps, $\mathcal{G}$, and $u_g(\mathbf{k})$) and $U'(\mathbf{k}, t)$ (with $T'$, relevant gaps, $\mathcal{G}$, and $u'(\mathbf{k})$). The two FGUs $U(\mathbf{k}, t)$ and $U'(\mathbf{k}, t)$ are defined to be topologically equivalent under the crystalline symmetry group $\mathcal{G}$ iff there exists a continuous deformation that connects them, preserves $\mathcal{G}$ and preserves all relevant gaps. As long as the crystalline symmetry group $\mathcal{G}$ for the topological equivalence is specified, we may refer to "topologically equivalent under $\mathcal{G}$" as "topologically equivalent" in short. Suppose we have two Floquet crystals $\hat{U}(t)$ (with $T$, a relevant gap choice, and $\mathcal{G}$) and $\hat{U}'(t)$ (with $T'$, a relevant gap choice, and $\mathcal{G}$). The two Floquet crystals $\hat{U}(t)$ and $\hat{U}'(t)$ are defined to be topologically equivalent iff there exists a continuous deformation that connects them, preserves $\mathcal{G}$ and preserves all relevant gaps. If two Floquet crystals are topologically equivalent, they must have topologically equivalent FGUs for any bases choices. Therefore, the topological distinction among FGUs must infer the topological distinction among the underlying Floquet crystals, and all topological invariants of FGUs can be applied to Floquet crystals. For this reason, we focus on the FGUs in this work.

The defined topological equivalence for FGUs is similar to the definition in Sec. 2 of ref. [68], except the following two differences. First, the definition in this work allows the deformation to deviate from the topologically equivalent FGUs by gauge transformations so that the defined topological equivalence is gauge invariant. Second, the definition in this work allows the symmetry representation and time period to vary along the deformation, and also allows the symmetry representation to depend on momenta. Furthermore, the topological classification based on the definition in this work may be different from the classification in refs. [69–71]. The topological equivalence defined in this work is immune to any global energy shift, while the same global energy shift may change value of the topological invariant in refs. [69–71].

Last but not least, a static limit is a Floquet crystal with static Hamiltonian; a static FGU is a FGU with static matrix Hamiltonian. A Floquet crystal (a FGU) with $\mathcal{G}$ is defined to have obstruction to static limits iff it is topologically distinct from all static limits (static FGUs) with $\mathcal{G}$.

**Return map**. The return map for the 1+1D $U(k, t)$ is constructed as follows. We first expand $U(k, T)$ as

$$U(k, T) = \sum_{m=1}^{2} e^{-i\mathcal{E}_{m,k}T} P_{k,m}(T), \tag{27}$$

where $P_{k,m}(T)$ is the projection matrix given by the eigenvector of $U(k, T)$ for $e^{-i\mathcal{E}_{m,k}T}$. With the above expression, the return map reads

$$U_\epsilon(k, t) = U(k, t)[U(k, T)]_\epsilon^{-t/T}, \tag{28}$$

where

$$[U(k, T)]_\epsilon^{-t/T} = \sum_{m=1}^{2} \exp\left[-\frac{t}{T}\log_{\epsilon_k}(e^{-i\mathcal{E}_{m,k}T})\right] P_{k,m}(T). \tag{29}$$

Here $\epsilon_k$ serves as the branch cut of the logarithm[69] by requiring $i\log_{\epsilon_k}(x) \in [\epsilon_k, \epsilon_k + 2\pi)$ for all $x \in U(1)$. As we always set the branch cut to be equal to the PBZ lower bound (i.e., $\epsilon = \Phi$), we have

$$i\log_{\epsilon_k = \Phi_k}(e^{-i\mathcal{E}_{m,k}T}) = \mathcal{E}_{m,k}T. \tag{30}$$

For the general situation, we just need to generalize the expression of the return map from the 1 + 1D two-band case to a $N$-band FGU $U(\mathbf{k}, t)$ with $T$. Specifically, we replace $k$ by $\mathbf{k}$ and replace 2 bands by $N$ bands in Eqs. (27)–(29) to get the return map

$$U_\epsilon(\mathbf{k}, t) = U(\mathbf{k}, t)[U(\mathbf{k}, T)]_\epsilon^{-t/T}, \tag{31}$$

where

$$[U(\mathbf{k}, T)]_e^{-t/T} = \sum_{m=1}^N \exp\left[-\frac{t}{T}\log_{\epsilon_\mathbf{k}}(e^{-i\mathcal{E}_{m,\mathbf{k}}T})\right]P_{\mathbf{k},m}(T), \quad (32)$$

and $P_{\mathbf{k},m}(T)$ is the projection matrix given by the eigenvector of $U(\mathbf{k}, T)$ for $e^{-i\mathcal{E}_{m,\mathbf{k}}T}$.

**Winding data and modulo operation**. The winding data of the 1+1D example is mathematically constructed as the follows. The return map commutes with the inversion symmetry representation at $k_0 = \Gamma/X$

$$u_\mathcal{P}(k_0)U_{\epsilon=\Phi}(k_0, t)u_\mathcal{P}^\dagger(k_0) = U_{\epsilon=\Phi}(k_0, t). \quad (33)$$

Combined with the representation of inversion symmetry in Eq. (1), the return map at $k_0$ has two blocks with opposite parties

$$U_{\epsilon=\Phi}(k_0, t) = \begin{pmatrix} U_{\epsilon=\Phi,k_0,+}(t) & \\ & U_{\epsilon=\Phi,k_0,-}(t) \end{pmatrix}. \quad (34)$$

Then we can define the following $U(1)$ winding number for each block

$$\nu_{k_0,\alpha} = \frac{\mathrm{i}}{2\pi}\int_0^T dt\,\mathrm{Tr}\left[U_{\epsilon=\Phi,k_0,\alpha}^\dagger(t)\partial_t U_{\epsilon=\Phi,k_0,\alpha}(t)\right] \in \mathbb{Z} \quad (35)$$

with $\alpha = \pm$ again labeling the parity. In particular, the integer-valued nature of $\nu_{k_0,\alpha}$ directly comes from time-periodic nature of the return map.

From the winding data $V$, a modulo operation $V \bmod \bar{A}$ is required to give the quotient winding data, as shown in Eq. (15). In practice, the modulo operation can be taken for the first nonzero component of $\bar{A}$ as discussed in the following. Eq. (16) shows that the first nonzero element of $\bar{A}$ is the its first element $\bar{A}_{\Gamma,+} = 1$, and then $V_Q = V + j\bar{A}$ with integer $j$ satisfying

$$V_{Q,\Gamma,+} = \nu_{\Gamma,+} + j\bar{A}_{\Gamma,+} = \nu_{\Gamma,+} \bmod \bar{A}_{\Gamma,+} = 0. \quad (36)$$

The generalization from the 1+1D example to a generic FGU is discussed in Supplementary Note 3.

**General framework**. In this part, we briefly introduce the general framework. Details can be found in Supplementary Note 3.

We consider a generic FGU with a generic crystalline symmetry group $\mathcal{G}$, and discuss its symmetry data first. The quasi-energy bands of the FGU are separated by relevant gaps into isolated sets of quasi-energy bands, and certain sets may contain more than one bands. Then, the symmetry contents, columns of the symmetry data, are defined for the isolated sets of quasi-energy bands. Each component of a symmetry content is the copy number of the corresponding irrep (like parity in the 1+1D example) of the little group at the corresponding high-symmetry momentum (like the inversion-invariant momentum in the 1+1D example). Nevertheless, the compatibility relation of all symmetry contents can always be expressed in terms of a compatibility matrix $\mathcal{C}$ just like Eq. (5), and all the symmetry contents belong to

$$\{BS\} \equiv \mathbb{N}^K \cap \ker\mathcal{C}, \quad (37)$$

where $\mathbb{N}$ is the set of non-negative integers. Here $K$ is the number of components of each symmetry content (which is 4 for the 1+1D example), and both $K$ and the compatibility matrix $\mathcal{C}$ can be determined solely based on $\mathcal{G}$. The PBZ-dependence of the symmetry data can still be removed by the equivalence among symmetry data defined in Results, and inequivalent symmetry data still infers topological distinction. Therefore, we can perform a topological classification for FGUs—also for Floquet crystals—solely based on the symmetry data, similar to what we did for static crystals.

Now we discuss the winding data of the generic FGU. The winding data is still a vector that consists of the winding numbers resolved by the high-symmetry momenta and irreps. We demonstrate that we can always choose the same set of high-symmetry momenta and irreps for the winding data and symmetry data, and the winding data always obeys the same compatibility relation as the symmetry content. In addition, if an irrep at certain momentum is missing, the corresponding winding number must be zero, which is an extra constraint imposed on the winding data by the symmetry data. We can always express this extra constraint in terms of a diagonal matrix $\mathcal{D}$ as

$$\mathcal{D}V = 0. \quad (38)$$

Then, the winding data takes value from the following group $\{V\}$

$$\{V\} \equiv \mathbb{Z}^K \cap \ker\mathcal{C} \cap \ker\mathcal{D}. \quad (39)$$

In the $1 + 1D$ example (Eq. (4)), all inequivalent irreps appear at all high-symmetry momenta, resulting in $\mathcal{D} = 0$ and Eq. (12). Similar to the $1 + 1D$ example, a generic FGU also has an infinite number of winding data, given by varying PBZ.

To solve the infinity issue, we construct the quotient winding data. For the generic FGU, the quotient winding data is still defined as Eq. (15), but $\bar{A}$ needs to be chosen carefully as discussed below. In the 1+1D example, $\bar{A}$ is the sum of all columns of the symmetry data just because all PBZ shifts that keep the symmetry data are (or are equivalent to) the $2\pi n$-shifts of the PBZ. For the generic FGU with in total $L$ isolated sets of quasi-energy bands, $2\pi$-shift of the PBZ is equivalent to

shifting the PBZ lower bound through $L$ isolated sets, which certainly leaves the symmetry data invariant. Nevertheless, in certain cases, the symmetry data is kept invariant even if we shift the PBZ lower bound through $0 < \tilde{L} < L$ isolated sets, and then we should choose $\bar{A} = \sum_{l=1}^{L_{KSD}} A_l$ for the construction of the quotient winding data, where $L_{KSD}$ is the smallest $\tilde{L}$ and $A_l$'s are columns of the symmetry data. After choosing proper $\bar{A}$, the equivalence between quotient winding data defined in Results still holds in the general framework.

Now let us turn to the DSI for the generic FGU. As mentioned in Results, we need to use Hilbert bases, which will be discussed with more details below. As shown in Eq. (37), the symmetry contents compatible with $\mathcal{G}$ always take value from the set $\{BS\}$. Mathematically speaking, $\{BS\}$ is a monoid rather than a group, since the components of a symmetry content are always non-negative, preventing nonzero elements in $\{BS\}$ from having inverse. We call a nonzero element in $\{BS\}$ irreducible[99] if it cannot be expressed as the sum of any two other elements in $\{BS\}$; otherwise, it is called reducible. In particular, the irreducible symmetry contents form a unique set of bases of $\{BS\}$[27,99], called the Hilbert bases, which we label as $a_i$ with $i = 1, 2, \ldots, I$.

As discussed in Results, the symmetry data can be classified as irreducible or reducible. We first discuss the DSI set for generic FGUs with irreducible symmetry data. When the symmetry data is irreducible (like the 1+1D example), the symmetry data is spanned by a unique set of Hilbert bases $\{a_j\}$ with $j$ taking $J$ different values in $\{1, 2, \ldots, I\}$. In this case, the static winding data set $\{V_{SL}\}$ for constructing the DSI set $\mathcal{X}$ simply reads

$$\{V_{SL}\} = \left\{\sum_j a_j q_j | q_j \in \mathbb{Z}\right\}. \quad (40)$$

From $\{a_j\}$, we can also determine the $K$ (as the number of components of $a_j$) and $\mathcal{D}$ (as shown in Supplementary Note 3) in Eq. (39). Then, combined with compatibility matrix $\mathcal{C}$, $\mathcal{X}$ can be directly derived based on Eq. (39) and the first equality in Eq. (21), meaning that $\mathcal{X}$ is uniquely determined by $\{a_j\}$ and $\mathcal{C}$. In particular, if two FGUs have the same $\mathcal{G}$ and have irreducible symmetry data that involve the same set of Hilbert bases, they have the same $\mathcal{X}$, no matter whether the two FGUs have equivalent symmetry data. As mentioned in Results, this simplification allows us to enumerate all possible DSI sets for irreducible symmetry data by considering all $2^I - 1$ nontrivial combinations of Hilbert bases of a given crystalline symmetry group $\mathcal{G}$.

When the symmetry data is reducible, then it is possible that more than one sets of Hilbert bases can span the symmetry data. Nevertheless, the DSI set in this case can be constructed from the tensor product of the DSI sets for irreducible symmetry data. Therefore, the DSI sets for irreducible symmetry data serve as the elementary building blocks for all DSI sets.

## Data availability
The datasets generated during and/or analyzed during the current study are available from the corresponding author on reasonable request.

## Code availability
The Mathematica and SageMath code generated during and/or analyzed for the current study are available from the corresponding author on reasonable request.

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

## Acknowledgements

We thank Yu-An Chen, Yang Ge, Biao Huang, Biao Lian, Xiao-Qi Sun, Jian-Xiao Zhang, Junyi Zhang, and in particular Sankar Das Sarma and Zhi-Cheng Yang for helpful discussions. J.Y. and R.-X.Z. are supported by the Laboratory for Physical Sciences. J.Y. acknowledges the UMD Libraries' Open Access Publishing Fund. R.-X.Z. acknowledges a JQI postdoctoral fellowship. Z.-D.S. is supported by the DOE Grant No. DE-SC0016239.

## Author contributions

J.Y. conceived the project. J.Y. performed the theoretical analysis with the input from R.-X.Z. and Z.-D.S. J.Y., R.-X.Z. and Z.-D.S. wrote the manuscript.

## Competing interests

The authors declare no competing interests.
