## [Peer Review File · Nature Communications]

REVIEWER COMMENTS

Reviewer #1 (Remarks to the Author):

In this work, the authors address two questions regarding non-interacting Floquet phases of matter, i.e., system subjected to periodic driving. First, how to tell two Floquet systems are topologically distinct? Second, how to tell if a Floquet system cannot be realized statically. The authors focus on Floquet systems with discrete time translational symmetry and static crystalline symmetry but no time reversal, particle-hole, or chiral symmetries. To study the first question, they provide a general framework based on the symmetry data of the static crystal and the quotient winding number at the high symmetry momentum in the Brillouin zone. Different symmetry data or quotient winding numbers indicate distinct Floquet phases, but not vice versa. To study the second question, the authors introduce the dynamical symmetry indicator (DSI) from the winding number. If the DSI of a Floquet system is different from all possible static systems that share the same symmetry data, then this Floquet system has obstruction to the static limit. This classification scheme is an extension of the symmetry indicator scheme in classifying static band structure. Based on this classification scheme, the authors enumerate nontrivial DSI for all 2D crystalline symmetries.

Floquet engineering and Floquet phase of matter is an important field. It is well-known that Floquet systems can host anomalous phases of matter that do not have static counterparts. Understanding Floquet phase of matter that has obstruction to the static limit provides valuable lessons on searching for exotic dynamical phenomena and realizing non-equilibrium phases of matter. However, in my opinion, the current manuscript is too formal and lacks a clear interesting physical result. It is not discussed in the manuscript whether this new classification scheme can lead to new dynamical phenomena, and, if it is the case, what they are. In addition, the classification scheme is not complete. Floquet systems with the same symmetry data/winding number may still be topologically distinct. It is not clear how some well-known examples of anomalous non-interacting Floquet phases of matter, such as the anomalous edge state which is characterized by a different winding number than that used in the work (ref[64]), fit into the classification scheme. I do not recommend publication at this stage. The significance of the work can be enhanced if new dynamical phenomena are identified based on the classification scheme.

Reviewer #2 (Remarks to the Author):

In this work, Yu et al develop a theory for efficiently diagnosing topological distinctions between non-interacting Floquet crystals and identifying the presence of an obstruction to a static limit. The theory builds upon the framework of symmetry indicators originally developed for diagnosing static topological phases. An important new ingredient introduced by the authors is the use of symmetry-resolved winding data, which can capture the nontrivial dynamics of the micro-motion within a Floquet period. Although both the symmetry and winding data suffer from ambiguity stemming from that in defining the PBZ, the authors provide an explicit way to quotient out the ambiguity and retain only the physical information. Among the possible winding data one could then quotient out those consistent with a static system and derive indicators for intrinsic Floquet phases.

The paper is generally well-written and the framework developed is thorough. In light of the recent interest in realizing topological phenomena in metamaterial and cold-atom platforms, the present work is likely helpful to the many groups worldwide working on related problems.

While I recommend publishing the manuscript on Nature Communications, I'd also suggest that the

authors should expand on the last paragraphs and elaborate further on how their theory could be related to experimental observables. It will help the readers to appreciate the significance of their work if the authors can provide some more discussion on the physical observables one could expect from the phases with nonzero DSI. Currently the authors mentioned two points: (i) symmetry breaking or gap closing in connecting a system with nonzero DSI to the static limit, and (ii) the suggested connection to mod 2 Chern number for the p2 plane group. Both of these points could be substantiated. For instance, what is the expected pattern of gap closing, e.g., does it have to be happening at all quasi-energy gaps? Can it be observed by just looking at the quasi-energy spectrum of the Floquet unitary, which is not obviously carrying information about the micro-motion? Similarly, for point (ii), is there any other possible boundary signatures other than the connection to Chern number for a general space group?

Lastly, a few typos spotted:

- Last paragraph under "Results": "topologically distinction"
- Before Eq. (10), "resulting a integer-valued"
- Paragraph under Eq. (21), "we call ... have equivalent (inequivalent) quotient winding data"
- Under Eq. (26), "based on the first quality in Eq. (24)"

Note: In the following, we cite all references according to the updated manuscript.

Reviewer #1 (Remarks to the Author):

In this work, the authors address two questions regarding non-interacting Floquet phases of matter, i.e., system subjected to periodic driving. First, how to tell two Floquet systems are topologically distinct? Second, how to tell if a Floquet system cannot be realized statically. The authors focus on Floquet systems with discrete time translational symmetry and static crystalline symmetry but no time reversal, particle-hole, or chiral symmetries. To study the first question, they provide a general framework based on the symmetry data of the static crystal and the quotient winding number at the high symmetry momentum in the Brillouin zone. Different symmetry data or quotient winding numbers indicate distinct Floquet phase, but not vice versa. To study the second question, the authors introduce the dynamical symmetry indicator (DSI) from the winding number. If the DSI of a Floquet system is different from all possible static systems that share the same symmetry data, then this Floquet system has obstruction to the static limit. This classification scheme is an extension of the symmetry indicator scheme in classifying static band structure. Based on this classification scheme, the authors enumerate nontrivial DSI for all 2D crystalline symmetries.

Reply: We thank the reviewer for the careful review of our work.

Floquet engineering and Floquet phase of matter is an important field. It is well-known that Floquet systems can host anomalous phases of matter that do not have static counterparts. Understanding Floquet phase of matter that has obstruction to the static limit provides valuable lessons on searching for exotic dynamical phenomena and realizing non-equilibrium phases of matter. However, in my opinion, the current manuscript is too formal and lacks a clear interesting physical result. It is not discussed in the manuscript whether this new classification scheme can lead to new dynamical phenomena, and, if it is the case, what they are.

Reply: We thank the reviewer for raising this important question that helps improve the significance of our work. The main concern of the reviewer is that the original version of our work did not demonstrate any new dynamical phases that can be directly derived from our theory. We

would like to emphasize that our theory can indeed lead to new predictions of dynamical topological matter. As a demonstration, we apply our scheme to the 3+1D inversion-invariant case ($P\bar{1}$ (#2) space group), and find that **the resultant classification predicts a new 3+1D anomalous Floquet second-order topological insulator (AFSOTI)**. Such exotic 3+1D AFSOTI features anomalous chiral hinge modes that are solely protected by spatial inversion symmetry, and has never been studied in the literature before.

Specifically, we find a variety of nontrivial DSI sets for the $P\bar{1}$ space group, including $\mathbb{Z}, \mathbb{Z}^2, \dots, \mathbb{Z}^6, \mathbb{Z}^7$, etc. Here \mathbb{Z}^n means that the DSI should be a n-dimensional vector with integer components. In principle, many of the phases with nonzero DSIs are new dynamical phases since the 3+1D dynamical phases protected by inversion symmetry have not been systematically discussed.

To explicitly demonstrate what the phases can be, we pick one DSI set \mathbb{Z}^7 that corresponds to a trivial symmetry data—symmetry data that is reproducible in atomic insulators—and we explicitly construct a 3+1D $P\bar{1}$ -invariant model that reproduces the trivial symmetry data. The model has four bulk quasi-energy bands, which are split into two isolated sets by two relevant gaps, one 0-gap and one π -gap. (See (a,b) of the following figure.) Direct calculation shows that the seven-dimensional DSI of this model equals to $(2,2,2,2,2,2,2)$, which is nonzero, indicating obstruction to static limits. As a result of the nontrivial dynamics characterized by the nonzero DSI, the system has chiral hinge modes in each bulk relevant gap. (See the following figure.)

Figure: **3+1D AFSOTI with anomalous chiral hinge modes.** This figure is a copy of Fig. 3 in the updated manuscript. In (a), we show an inversion-preserving configuration of the 3+1D AFSOTI that is finite along x, y and infinite along z . The purple and red lines indicate the hinges that host anomalous chiral hinge modes. In (b), we plot the quasi-energy band structure of the 3+1D AFSOTI for the configuration in (a). \mathcal{E} and T label the quasi-energy and time period, respectively. The gray lines are the bulk bands (as well as the dangling surface bands). “I.S.” labels isolated set of bulk quasi-energy bands, and each isolated set contains two bulk quasi-energy bands. “R.G.” stands for the bulk relevant gap, and there are two relevant gaps, one at $\mathcal{E}T = 0$ (called 0-gap) and the other at $\mathcal{E}T = \pi$ (called π -gap). The purple and red lines mark the chiral hinge modes localized at the purple and red hinges in (a), respectively. The orange dashed lines mark the boundary of the phase Brillouin zone (PBZ). In (c), we consider an inversion-preserving configuration of the 3+1D AFSOTI that is finite along all three spatial directions, and plot total probability density of the two eigenmodes with quasi-energies closest to 0 or $\pi \pmod{2\pi}$.

Now we explain why the chiral hinge modes are anomalous. For 3+1D static insulators with $P\bar{1}$ space group, the stable in-gap chiral hinge modes (for an inversion-invariant open boundary condition) only appear when the occupied bands have nontrivial π axion angle according to Ref. [100]. In other words, the static topological invariant for the chiral hinge modes is the inversion-protected axion angle, analogous to the Chern number as the static topological invariant for chiral edge modes. In our model, both isolated sets of quasi-energy bands have 0 axion angle, owing to the trivial symmetry data, meaning that chiral hinge modes in our model **cannot** originate from the static topology. Therefore, the chiral hinge modes must come from the nontrivial dynamics that is characterized by nonzero DSI, and are anomalous. In sum, our inversion-invariant model is a 3+1D AFSOTI that has anomalous chiral hinge modes in the absence of nonzero axion angles, and represents a truly anomalous phase without any static analogues, in analogy to the dynamical phase in Ref. [64] that has anomalous chiral edge modes in the absence of nonzero Chern numbers.

In particular, our model is the first 3+1D AFSOTI solely protected by the static crystalline symmetries. Although hinge modes in 3+1D Floquet insulators were discussed in Ref. [77] and Ref. [97], our model is fundamentally different from theirs. First, the hinge modes originate from the time glide symmetry in Ref. [77] and from the effective spectral symmetry in Ref. [97], both of which are not static crystalline symmetries, while our model is protected by static inversion symmetry. Second, there was no discussion on whether the bulk quasi-energy bands in Ref. [77] and Ref. [97] have trivial static topology, and therefore it is not clear whether the hinge modes in Ref. [77] and Ref. [97] have any static topological origin. Yet, the relevant static topological invariants in our model are confirmed to be trivial, and therefore the chiral hinge modes in our model must be anomalous. This difference is also supported by the fact that the hinge modes were only demonstrated in either the bulk π -gap (Ref. [77]) or the bulk 0-gap (Ref. [97]), while the hinge modes appear in both bulk 0-gap and π -gap in our model. Owing to these differences, we believe our model is the first 3+1D AFSOTI solely protected by the static crystalline symmetries, which is predicted by the inversion-protected \mathbb{Z}^7 DSI obtained above. Moreover, the DSI currently serves as the only topological invariant for the nontrivial dynamics, as well as the resultant the anomalous chiral hinge modes, in the model. We added a new section in the Results to discuss this point.

Besides the main concern, the reviewer also raised the concern that our manuscript could be too formal for general readers. In the revised manuscript, we have improved the readability of our work by (i) removing all the abstract discussions on the general framework in the Results and summarizing them in a new section of the Methods, and (ii) providing more discussions on the explicit models to demonstrate our formalism. Our revision is summarized in the “List of Changes” session below.

In addition, the classification scheme is not complete. Floquet systems with the same symmetry data/winding number may still be topologically distinct.

Reply: We completely agree with the reviewer that our classification is incomplete in the sense that two Floquet crystals with the same symmetry data and quotient winding data can still be topologically distinct. As we will explain below, however, the incompleteness of our classification does not spoil the novelty, importance, and impact of our work.

First, such incompleteness problem is a well-known intrinsic issue for all general classifications for static systems with nontrivial crystalline symmetries, including the famous topological quantum chemistry (Ref. [30]), symmetry indicator (Ref. [31]), and the K-theory classification (Ref. [101]). For example, for $p3$ plane group without time-reversal symmetry, the symmetry indicator provides a $\mathbb{Z}_3 = \mathbb{Z}/3\mathbb{Z}$ classification, but we know the classification is at least \mathbb{Z} from Chern number, meaning that phases with the same symmetry indicator can still have different nontrivial topology with different Chern numbers. Similar issue also exists for the topological quantum chemistry, as symmetry representations do not carry all topological information. Moreover, the K-theory simply misses all fragile topological phases, meaning that even if K-theory identifies certain phases as trivial, they can still have nontrivial fragile topology (Ref. [22]).

However, the incompleteness of these static theories does not jeopardize their importance at all, since (i) they were the first static classification schemes that are applicable to all crystalline symmetry groups, and (ii) the topological quantum chemistry and symmetry indicator are also very efficient as they only involve a small number of the momenta in the Brillouin zone. These features of the theories allowed the later prediction of new static topological materials. (Ref. [33-36]) Similarly, although our scheme is incomplete, it (i) serves as the first general Floquet classification

scheme that is applicable to all crystalline symmetry groups, (ii) is very efficient since it involves the same number of the momenta in the Brillouin zone as the topological quantum chemistry and symmetry indicator, and (iii) a new anomalous Floquet phase has been explicitly predicted according to the theory as discussed above. Before our work, there was no Floquet classification scheme that is applicable to all crystalline symmetry groups, and the previously existing topological indices have relatively low efficiency. Therefore, we believe the incompleteness of our theory does not influence the importance of our work.

In general, deriving a complete topological classification in the presence of nontrivial crystalline symmetries is extremely difficult and nontrivial—since more than ten years ago, countless efforts have been dedicated to find a complete static classification for all crystalline symmetry groups, but we have not succeeded. Thus, to make our scheme complete, or in general to find a complete classification for Floquet crystals with nontrivial crystalline symmetries, is a very important future direction. We added a discussion on this future direction in the “Conclusion and Discussion” section of the updated manuscript.

It is not clear how some well-known examples of anomalous non-interacting Floquet phases of matter, such as the anomalous edge state which characterized by a different winding number than that used in the work (ref[64]), fit into the classification scheme.

Reply: We thank the reviewer for raising this important point. In systems with only lattice translations, DSIs typically cannot indicate the dynamical phases characterized by the winding number W defined in Ref. [64]. Nevertheless, we conjecture that in the presence of crystalline symmetries other than lattice translations, certain components of the DSI may equal to the winding number W defined in Ref. [64] (perhaps up to certain modulo operations owing to the quotient operation taken in the construction of the DSI). When this happens, the DSI provides an efficient way of indicating the nontrivial dynamics characterized by the winding number W , as the DSI only involves a small number of momenta while the winding number W requires the information over the entire 2D first Brillouin zone. The general proof of the conjecture is highly nontrivial, which we leave as a future work; nevertheless, we provide an explicit model that supports the conjecture in the following.

We construct a 2+1D dynamical tight-binding model with two-fold rotation symmetry (plane group $p2$). Our model is in the anomalous phase proposed in Ref. [64], because (i) our model has two bulk quasi-energy bands with zero Chern numbers, (ii) has one anomalous chiral edge mode at each edge in each bulk quasi-energy gap, and (iii) has the winding number W being 1. There is one difference between our model and the discussion in Ref. [64]: our model has a two-fold rotation symmetry, which is not always present in the model of Ref. [64]. In the presence of the two-fold rotation symmetry, DSI may take nonzero values as shown in Tab. I, and we find that in our specific model, all components of the DSI take the same values as the winding number W . Therefore, our model suggests that in the presence of the two-fold rotation symmetry, certain components of the DSI can equal to the winding number W defined in Ref. [64] in certain phases. Furthermore, in this case of plane group $p2$, the evaluation of DSI only involves four rotation-invariant momenta, which is much more efficient than the evaluation of the winding number W that involves the whole first Brillouin zone. We added the discussion on this model in the “2+1D DSI” section of the Results in the updated manuscript.

I do not recommend publication at this stage. The significance of the work can be enhanced if new dynamical phenomena are identified based on the classification scheme.

Reply: We hope our responses, especially the proposal of the new 3+1D AFSOTI with anomalous chiral hinge modes, are satisfactory for the reviewer.

Reviewer #2 (Remarks to the Author):

In this work, Yu et al develop a theory for efficiently diagnosing topological distinctions between non-interacting Floquet crystals and identifying the presence of an obstruction to a static limit. The theory builds upon the framework of symmetry indicators originally developed for diagnosing static topological phases. An important new ingredient introduced by the authors is the use of symmetry-resolved winding data, which can capture the nontrivial dynamics of the micro-motion within a Floquet period. Although both the symmetry and winding data suffer from ambiguity stemming from that in defining the PBZ, the authors provide an explicit way to quotient out the ambiguity and retain only the physical information. Among the possible winding data one could then quotient out those consistent with a static system and derive indicators for intrinsic Floquet phases.

The paper is generally well-written and the framework developed is thorough. In light of the recent interest in realizing topological phenomena in metamaterial and cold-atom platforms, the present work is likely helpful to the many groups worldwide working on related problems.

Reply: We thank the reviewer for the careful review of our work. We are grateful to the reviewer for stating that our paper is well-written, thorough, and likely helpful.

While I recommend publishing the manuscript on Nature Communications, I'd also suggest that the authors should expand on the last paragraphs and elaborate further on how their theory could be related to experimental observables. It will help the readers to appreciate the significance of their work if the authors can provide some more discussion on the physical observables one could expect from the phases with nonzero DSI. Currently the authors mentioned two points: (i) symmetry breaking or gap closing in connecting a system with nonzero DSI to the static limit, and (ii) the suggested connection to mod 2 Chern number for the $p2$ plane group. Both of these points could be substantiated.

Reply: We thank the reviewer for the helpful suggestions and for raising various important questions. In the following, we provide a point-to-point reply to the questions.

For instance, what is the expected pattern of gap closing, e.g., does it have to be happening at all quasi-energy gaps?

Reply: The gap closing is only required to happen in at least one of the relevant bulk quasi-energy gaps, and is not required to happen at all bulk quasi-energy gaps. For example, in the 1+1D example, both bulk quasi-energy gaps are treated to be relevant. Then, if connecting the 1+1D dynamical example to a static limit, the gap closing must happen in at least one of the two bulk quasi-energy gaps. Specifically, depending on the static limit to which the 1+1D dynamical system connects, the gap closing can happen (i) in only the first gap, (ii) in only the second gap, or (iii) in both gaps. In addition, if we choose only part of the bulk quasi-energy gaps to be relevant, then the gap closing is not required to happen in any of the remaining irrelevant gaps, i.e., the connection to static limit can happen without closing any irrelevant gaps. We added the discussion on this point in the “Conclusion and Discussion” section of the updated manuscript.

Can it be observed by just looking at the quasi-energy spectrum of the Floquet unitary, which is not obviously carrying information about the micro-motion?

Reply: It can be observed by just looking at the bulk quasi-energy spectrum of the time-evolution operator at the end of the time period. We agree that the bulk quasi-energy spectrum at specific values of the system parameters does not contain the essential information about the quantum dynamics, since the bulk quasi-energy spectrum can be reproduced by a static Hamiltonian in this case. However, the gap closing in the quasi-energy spectrum during the deformation of the system (or during the change of the parameter values) does carry the topological information about the micro-motion. The intuitive reasoning is presented in the following.

To deform the quasi-energy spectrum, one can perform two deformations: (i) continuously deforming the Floquet Hamiltonian given by the time-evolution operator at the end of the time period without caring about any other time moments, and (ii) continuously deforming the time-evolution operator over the entire time period. In the study of Floquet topology, what we perform is the deformation (ii), while the deformation (i) is for the static topology. Compared to the static deformation (i), the Floquet deformation (ii) would impose more constraints on the deformation of the quasi-energy spectrum, since the entire time period rather than one specific time moment is

included in the Floquet deformation. Such extra constraints make the deformation of the quasi-energy spectrum, as well as the quasi-energy gap closing during it, carry the information of the Floquet dynamics.

In fact, the gap closing in the bulk quasi-energy spectrum lies at the heart of the definition of the Floquet topology. Explicitly, two Floquet crystals are defined to be topologically distinct if their time-evolution operators can be continuously deformed into each other without closing any relevant bulk quasi-energy gaps and without breaking any symmetries. The brief version of this definition was stated in Ref. [68], and we made the definition precise in our work in order to develop a general framework. The definition is applicable to various anomalous topological phases, including the phase with anomalous edge modes proposed in Ref. [64], and has been used to explain experiments like Ref. [72]. We added the discussion on this point in the “Conclusion and Discussion” section of the updated manuscript.

Similarly, for point

(ii), is there any other possible boundary signatures other than the connection to Chern number for a general space group?

Reply: Yes. As discussed in the reply to the first reviewer, we propose a 3+1D inversion-protected AFSOTI that hosts anomalous chiral hinge modes. In this system, the inversion-protected DSI indicates the existence of anomalous chiral hinge modes. Such correspondence between the inversion-protected DSI and the anomalous chiral hinge modes is analogous to the correspondence between the inversion-protected static symmetry indicator and the static chiral hinge modes in static axion insulators. In the static case, the bulk-boundary correspondence of the inversion-protected symmetry indicator is established based on the fact that (i) the axion angle can be quantized by the inversion symmetry, and (ii) nonzero quantized axion angle can indicate the chiral hinge modes. (See Ref. [100].) Therefore, our model suggests that in the presence of inversion symmetry, the 3+1D DSI might have a bulk-boundary correspondence that is analogous to the bulk-boundary correspondence of the axion angle. As the axion angle is quantized not only by inversion but also by any improper space-group operation (like mirror, rotoinversion, etc), 3+1D DSIs might indicate anomalous chiral hinge modes for hundreds of space groups that contain at least one improper space-group operation. The general proof for such axion-like correspondence

is an interesting and important future direction. We added the discussion on this point in the “Conclusion and Discussion” section of the updated manuscript.

Lastly, a few typos spotted:

- Last paragraph under “Results”: “topologically distinction”
- Before Eq. (10), “resulting a integer-valued”
- Paragraph under Eq. (21), “we call ... have equivalent (inequivalent) quotient winding data”
- Under Eq. (26), “based on the first quality in Eq. (24)”

Reply: We thank the reviewer for pointing out these typos. We have fixed them in the updated manuscript.

REVIEWERS' COMMENTS

Reviewer #1 (Remarks to the Author):

In this revised version, the authors made significant improvements to the manuscript. In particular, they identified a new 3+1D anomalous Floquet second-order topological insulator based on their classification scheme. They also discussed the relationship between their classification scheme and the winding number used in ref 64 based on a specific example. All of my questions and comments were addressed. Therefore I recommend publication in nature communication.

Reviewer #2 (Remarks to the Author):

I have read through the initial referee reports, the authors' rebuttal, as well as the revised manuscript. The authors have satisfactorily address all my comments and the revised manuscript has been substantiated improved with more explicit connection to the physical properties of the proposed Floquet phases. As such, I recommend publication of the manuscript in Nature Communications in the current form.

Reviewer #1 (Remarks to the Author):

In this revised version, the authors made significant improvements to the manuscript. In particular, they identified a new 3+1D anomalous Floquet second-order topological insulator based on their classification scheme. They also discussed the relationship between their classification scheme and the winding number used in ref 64 based on a specific example. All of my questions and comments were addressed. Therefore I recommend publication in nature communication.

Reply: We thank the reviewer for the careful review of our paper and for the recommendation of the publication.

Reviewer #2 (Remarks to the Author):

I have read through the initial referee reports, the authors' rebuttal, as well as the revised manuscript. The authors have satisfactorily address all my comments and the revised manuscript has been substantiated improved with more explicit connection to the physical properties of the proposed Floquet phases. As such, I recommend publication of the manuscript in Nature Communications in the current form.

Reply: We thank the reviewer for the careful review of our paper and for the recommendation of the publication.